# Production Contracts and Food Quality: A Transaction Cost Analysis for the Italian Durum Wheat Sector

Angelo Frascarelli [1], Stefano Ciliberti [1,*], Gustavo Magalhães de Oliveira [2], Gabriele Chiodini [1] and Gaetano Martino [1]

1   Department of Agricultural, Food and Environmental Sciences, University of Perugia, 06121 Perugia, Italy; angelo.frascarelli@unipg.it (A.F.); gabriele.chiodini@unipg.it (G.C.); gaetano.martino@unipg.it (G.M.)

2   Center for Organization Studies (CORS), School of Economics, Business and Accounting (FEA), University of São Paulo (USP), Sao Paulo 05508-011, Brazil; gustavomoliv@gmail.com

*   Correspondence: stefano.ciliberti@unipg.it; Tel.: +39-075-585-7137

**Abstract:** Agribusiness firms requiring a consistent supply of high quality agricultural raw materials have increasingly adopted production contracts to coordinate their supply chains. The present work is aimed to shed light on the role played by sources of asset specificity and uncertainty related to quality strategies in the diffusion of contractual arrangements within the Italian durum wheat sector. To this purpose, factor analyses and probit regressions are estimated in data collected among durum wheat producers. The findings confirm that the role of asset specificity is negligible in presence of staple crops. Moreover, they reveal that experience, transparency and technological stability are all relevant aspects that reduce uncertainty and, in turn, trigger the adoption of production contracts as a governance solution for durum wheat.

**Keywords:** contracts; TCE; quality; uncertainty; agri-food; wheat; Italy

## 1. Introduction

The credence attributes of food have received an increasing attention due to several scandals in the agri-food sector [1]. In response to both consumers' demand for quality and safe products and to these stricter rules, the need for new types of organizational arrangements and governance solutions emerged to face information asymmetries and opportunistic behaviors [2–5]. Accordingly, agricultural and food regulations and private strategies have triggered the diffusion of specific standards [2,6]. These new standards, in turn, have raised relevant organizational issues, shedding light on the need of investigating chain coordination solutions as a key element for food quality strategies [7].

In the Italian production of pasta, contracts have been progressively diffused among farmers [8]. Contractual solutions have gained momentum to trace and to promote high-quality productions [9,10]. Production contracts can entail a broad variety of incentive instruments, ranging from input control field visits and quality assessment to premium prices [11]. However, there is still lack of understanding on which factors lead to the diffusion of contracts along the durum wheat supply chain. Investigating the determinants toward contract adoption might bring relevant implications. To this aim, this study adopts the transaction cost economics (TCE) approach [12]. At the heart of TCE, contracts are seen as a modality of governance operating differently among alternative organizational solutions so as to yield a transaction cost economizing [13].

In more detail, the paper investigates whether and how the increasing demand for quality in the agri-food supply chain triggers the diffusion of contractual arrangements. Here, we focus on asset specificity and uncertainty for investigating such a phenomenon.

The remainder of the paper is organized as follows. The conceptual framework analyzes the role of quality requirements in fostering coordination by means of contractual solutions, according to the TCE perspective. Section 2 presents research hypotheses.

The source of data and empirical analysis are described in Sections 3 and 4. Then, results are illustrated and discussed. Lastly, the paper ends with some final remarks.

## 2. Conceptual Framework

### 2.1. Food Quality, Coordination and Contracts

The agri-food industry has remarkably changed over the last few decades, since coordination issues related to food quality have escalated in importance [14]. Food quality encompasses a bunch of complex attributes as " ... It requires tight coordination among transactors, with respect to the definition of detailed standards, methods of production and controls for guaranteeing the conformity of products to what is signaled ... " [1] (p. 427). As pointed out by Masten [15], different characteristics in agricultural production must be taken into account when investigating food quality: although many agricultural products are subject to strong climatic uncertainty and perishability constraints that influences quality, contracts can settle ex ante terms and conditions by which quality variation is coordinated.

Quality may be recognized as an important driver of the diffusion of contracts in agri-food chains. Goodhue [16] extensively reviewed the literature on information asymmetry and food quality and investigated the role of contracts in food quality supply. Agri-food chains provide an elucidative benchmarking for evaluating the role of contracts in providing food quality [16]. Contractual arrangements are coordination forms that can minimize free rider and opportunistic behaviors due to the possible use of third-party arbitrations in case of disagreements [7,17]. In fact, contracts may reduce transaction costs, trigger specific investment influencing food quality and facilitate risk-sharing among chain agents [4,18–20].

This study assumes that the management of coordination problems addressed by contracts in food chains must specify contractual terms determining how farmers contribute to quality strategies. In fact, although always incomplete [21], contracts reveal how buyers remunerate farmers, as well as which restrictions farmers must follow and respect, such as the set of inputs (seeds, pesticides, fertilizers) allowed to be used, minimum requirements or a range of acceptable values for specific quality attributes or financial incentives for improved quality. In this latter case, contracts establish conditions and agreements between parties, thereby indicating specific activities that farmers must invest in and specifying which parameters would be used by buyers to remunerate investments aimed to pursue quality strategies [22–25].

### 2.2. The Role of Transactional Attributes in the Durum Wheat Sector

TCE states that coordination issues mainly derive from asset specificity and uncertainty. Asset specificity refers to specific investments that might lost a great share of value when applied in an alternative purpose rather the original one [12]. Uncertainty emerges as the inability of predict all possible events that could (or not) happen in a particular context [12].

The role of asset specificity related to quality recently emerged in the pasta supply chain. Producers and processors started to make large investments on product differentiation through the adoption of brands related to particular varieties of durum wheat (e.g., traditional, local) and/or specific schemes of certified quality (e.g., organic). Italian companies have launched pasta produced by using 100% Italian grains semolina brands, often by recovering traditional local varieties to seize new market opportunities [9]. Due to the request for high quality durum wheat, farmers have become portfolios of transaction-specific skills that entail the uptake of technical assistance services [26]. This new scenario, aimed to ensure quality of production, therefore, becomes sources of asset specificities [1].

In addition, adopting a specific technology in a framework of extreme volatility has generated uncertainty about the cost/benefit of existing alternatives [27]. The need for adaptation to unpredictable technologies pushed parties to pool rights within contracts in order to reduce misalignment costs [28]. Second, quality increased bilateral dependency

among trading partners opening room for contractual hazard [29]. Since agricultural products were increasingly valued for specific attributes and for the practices used in their production, sellers could exploit their informational advantage to cheat on quality performance achieved [7]. As a consequence, this has forced millers to strictly monitor farmers, who may act opportunistically [30] and it has led to more centralized governance structures in modern food chains [15,19].

Due to the emerging coordination imposed by food quality requirements, we expect that both asset specificity and uncertainty are relevant transaction attributes playing a role in the diffusion of contracts in the durum wheat sector (Figure 1).

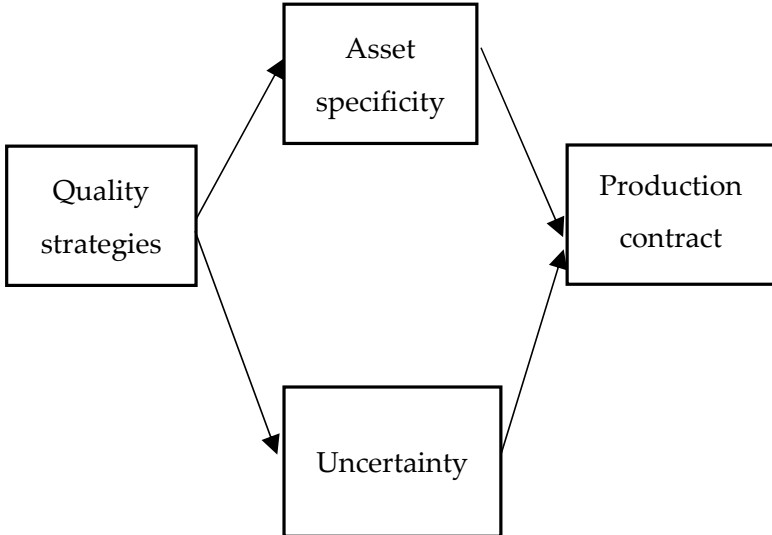

**Figure 1.** Conceptual model.

Accordingly, two hypotheses are elaborated as follows:

**Hypothesis 1 (H1).** *The presence of asset specificity in durum wheat production affects the likelihood to adopt production contracts.*

**Hypothesis 2 (H2).** *The presence of uncertainty in durum wheat production affects the likelihood to adopt production contracts.*

## 3. Material and Methods

*3.1. Data Collecting: Questionnaire and Variables*

Data were gathered by face-to-face interviews with durum wheat producers during two technical conferences held in Southern Italy—where the Italian production of durum wheat is mainly concentrated—in fall, 2015. Before each event, organizers were contacted in order to authorize the collection of questionnaires, then farmers were requested to participate just on the day of the conference. The resulting sample is a non-probabilistic convenience sample.

The questionnaire (see Supplementary Material Annex S1) is composed of several sections. In addition, to questions aimed to get information about the characteristics of the respondents, such as age, level of education (first section) and to test the level of knowledge about the durum wheat sector (second section), the analysis mainly focused on aspects related to trade relationships with other firms (third section). They are related to the use of production contracts, the participation in cooperatives and farmers' perceptions on the elements that affect the choice of the trading partners in order to sell durum wheat. In more detail, we focused on uncertainty and asset specificity. These two constructs entail different aspects which concern to behavioral, market and technological uncertainty [12,27] and different types of asset [31]. Accordingly, we operationalized these concepts by different

variables. As for the uncertainty, we concentrated on difficulty to predict the technological change [32], the intensity of the processor inducements to adopt a given technology [33,34], the transparency and the easiness to understand the content of the contractual terms [18,35]. Therefore, the farmers were required to evaluate 5-point Likert scale questions (see Table 1 for descriptive statistics) related to the relevance of the

- stability of the technology required in the contracts (STAB_TECH);
- technological change pressing the company (PRESS_TECH);
- transparency related to the contractual objectives (GOAL);
- clarity of contractual clauses referred to quality standards (INDEX);
- farmers' experience in foreseeing contractual objectives (EXP).

As for the asset specificity we directly focused on the contingencies identified by Williamson [31]. Again, respondents were required to evaluate 5-point Likert scale questions that encompass motivations affecting their transactions of durum wheat sale, such as

- localization in specific and favorable geographic area (ZONE);
- presence of specific physical resources (i.e., processing plants) (RES);
- high skilled staff with a high level of expertise (STAFF);
- presence of specific investments (i.e., transportation means, stores, warehouses and so on) (INVEST);
- presence of commercial brands (BRAND).

**Table 1.** Variables and descriptive statistics.

| Variable | Code | Scale of Measurements | N. obs | Rel. fr. % |
|---|---|---|---|---|
| Use of contracts | C | 0: no<br>1: yes | 150 | 59.33<br>40.67 |
| Relevance of specific physical resources in affecting the choice of the trading partner | Res | 1: strongly disagree<br>2: disagree<br>3: neither agree not disagree<br>4: agree<br>5: strongly agree | 146 | 5.48<br>9.59<br>10.96<br>33.56<br>40.41 |
| Relevance of dedicated investments in affecting the choice of the partner | INVEST | 1: strongly disagree<br>2: disagree<br>3: neither agree not disagree<br>4: agree<br>5: strongly agree | 145 | 7.59<br>8.28<br>11.72<br>35.17<br>37.24 |
| Relevance of specific geographical location in affecting the choice of the partner | Zone | 1: strongly disagree<br>2: disagree<br>3: neither agree not disagree<br>4: agree<br>5: strongly agree | 144 | 6.25<br>7.64<br>7.64<br>32.64<br>45.83 |
| Relevance of human capital in affecting the choice of the partner | STAFF | 1: strongly disagree<br>2: disagree<br>3: neither agree not disagree<br>4: agree<br>5: strongly agree | 144 | 6.25<br>9.03<br>6.25<br>46.53<br>31.94 |
| Relevance of commercial brand in affecting the choice of the partner | Brand | 1: strongly disagree<br>2: disagree<br>3: neither agree not disagree<br>4: agree<br>5: strongly agree | 146 | 14.38<br>4.79<br>23.97<br>23.29<br>33.56 |

**Table 1.** *Cont.*

| Variable | Code | Scale of Measurements | N. obs | Rel. fr. % |
|---|---|---|---|---|
| Relevance of the technological pressure | Tech_press | 1: strongly disagree<br>2: disagree<br>3: neither agree not disagree<br>4: agree<br>5: strongly agree | 123 | 9.76<br>16.26<br>7.32<br>39.84<br>26.83 |
| Relevance of the technological stability | TECH_STAB | 1: strongly disagree<br>2: disagree<br>3: neither agree not disagree<br>4: agree<br>5: strongly agree | 127 | 12.60<br>14.17<br>13.39<br>40.16<br>19.69 |
| Relevance of the contractual transparency related to contractual objectives | GOAL | 1: strongly disagree<br>2: disagree<br>3: neither agree not disagree<br>4: agree<br>5: strongly agree | 125 | 14.40<br>16.80<br>11.20<br>28.00<br>29.60 |
| Relevance of the contractual clarity referred to quality standards | INDEX | 1: strongly disagree<br>2: disagree<br>3: neither agree not disagree<br>4: agree<br>5: strongly agree | 132 | 12.12<br>10.61<br>9.85<br>37.12<br>30.30 |
| Relevance of experience in predicting contractual objectives | Exp | 1: strongly disagree<br>2: disagree<br>3: neither agree not disagree<br>4: agree<br>5: strongly agree | 128 | 9.38<br>9.38<br>9.38<br>34.38<br>37.50 |
| Age | AGE_ n | 1: <30 years old<br>2: 30–50 years old<br>3: 50+ years old | 158 | 8.23<br>53.80<br>37.97 |
| Education level | EDU_n | 1: Elementary school<br>2: Middle school<br>3: High school<br>4: University | 158 | 1.90<br>15.82<br>63.92<br>18.35 |

### 3.2. Description of the Sample

Our sample consists of 158 observations. Respondents vary in age, education, membership of cooperatives and other characteristics. The sample is composed by 90% of farmers over 30 years (38% over 50 years) in line with the national trend for the agricultural sector in Italy, 64% consisting of farmers with high school education and 40% joining cooperatives that sells durum wheat.

Within the 40% of the sample that adopts production contract for durum wheat, 57 farmers provide details of the main contractual attributes they used to negotiate (Figure 2).

What emerged is that the contractual term referred to price and quality premium are largely predominant, followed by the rules of production and time of payment. Interestingly, in 32% and 26% of the above-mentioned 57 cases, contracts, respectively, contain one or two terms (referred to price or/and rules of production or/and quality premium), whereas three or four terms are less diffused (15% and 5% of the cases, respectively).

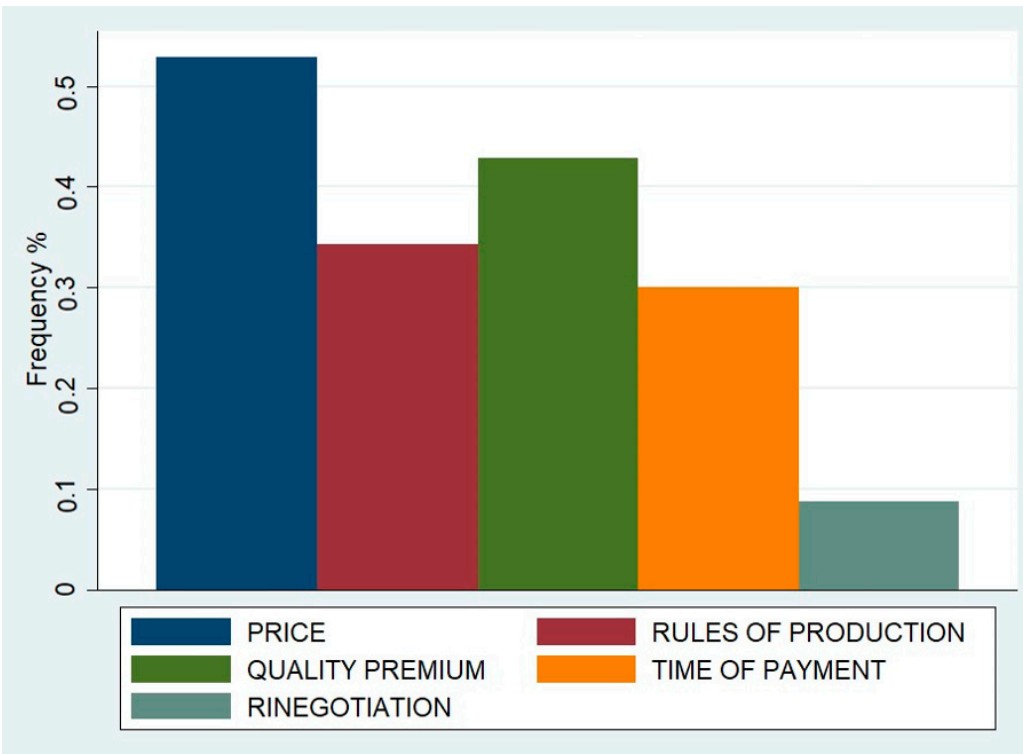

**Figure 2.** Frequency of the contractual terms used in production contracts (*n* = 57).

## 4. Data Analysis

### 4.1. Factor Analysis

We performed a factor analysis (FA) aimed at determining the number and nature of latent variables (attributes of transaction in our case) that explains the variation and covariation in a set of measured variables [36]. FA reduces the number of variables by describing linear combinations of the variables that contain most of the information and that could admit meaningful interpretations. FA finds a few common factors (say, k of them) that linearly reconstruct the *p* original variables:

$$y_{ij} = z_{i1}b_{1j} + z_{i2}b_{2j} + \cdots + z_{ik}b_{kj} + e_{ij}$$

where $y_{ij}$ is the value of the $i_{th}$ observation on the $j_{th}$ variable, $z_{ik}$ is the $i_{th}$ observation on the $k_{th}$ common factor, $b_{kj}$ is the set of linear coefficients called the factor loadings and $e_{ij}$ is similar to a residual but is known as the $j_{th}$ variable's unique factor. Once the factors and their loadings have been estimated, they are interpreted examining the $b_{kj}$'s and assigning names to each factor. In this regard, loadings represent the weights and correlations between each variable and the factor, so the higher the load the more relevant it is in defining the factor's dimensionality.

In this specific case, a factor analysis is used in order to explore whether and how some 5-points Likert scale variables could be grouped into a smaller number of factors, or constructs. This study includes only ordinal variables, then a polychoric factor analysis was performed according to Holgado-Tello et al. [37]. Based on Preacher and MacCallum [36], the following procedure is implemented: (i) combinations of three criteria (the Kaiser criterion, the subjective scree test and the parallel analysis) are used to determine the appropriate number of factors to retain, opting for two factors after attempts for three factors yielded very poor results, as indicated by the parallel test, (ii) oblique rotation methods are adopted instead of orthogonal rotation since factors are rarely if ever uncorrelated in empirical studies, (iii) all obtained factor loadings are reported.

### 4.2. Reduced Form Analysis

The second step of the empirical analysis focused on the relationship between the factors identified and the choice of a production contract as a governance structure organizing the transaction between the farmers and the processors.

According to the conventional TCE approach, due the difficulty of directly measuring transaction costs, the model focused on the direct impact of transactional attributes on the trade-off between alternative modality of governance among organizational solutions, assuming they do so by shaping the associated transaction costs [38]. Therefore, due to the categorical nature of the dependent variable, a traditional probit model with robust standard errors is implemented. This is a model for a binary dependent variable, assuming that the probability of a positive outcome is determined by the standard normal cumulative distribution function. The specification adopted in order to test the research model is the following:

$$C = \Phi \, (\text{ß}_0 + \text{ß}_1 \, \text{FACTOR}_1 + \text{ß}_n \, \text{FACTOR}_2 + \text{ß}_3 \, \text{AGE\_n} + \text{ß}_4 \, \text{EDU\_n})$$

whereas C is the dependent variables indicating the decision to use/not use contracts, $\text{FACTOR}_1$ and $\text{FACTOR}_2$ are the latent construct obtained from the factor analysis, representing asset specificity and absence of uncertainty-Lastly, AGE\_n and EDU\_n are categorical variables (see Table 1 for descriptive statistics) that are introduced to control for age and education levels [39,40]. The model is estimated with Stata 14.2, using the maximum likelihood procedure. Results for the specification error and goodness of fit tests (e.g., log pseudo likelihood $\chi 2$ and pseudo R2) and postestimation analysis are also provided in the results section.

## 5. Results

The outcomes of the factors analyses reveal that one indicator's communality (PRESS\_TECH) is lower than the precautionary threshold of 0.2 proposed by Child, D. [41]. As a result, this variable must be discarded and a new FA (according to the above-mentioned procedure) is performed with only 9 out of the 10 available variables. Again, a 2-factor FA with 9 variables is performed, whose results are reported in the right side of the Table 2.

**Table 2.** Polychoric 2-factors FAs loadings after oblique rotations (*) (**).

| Variable | FA (10 Variables) | | | FA (9 Variables) | | |
|---|---|---|---|---|---|---|
| | **Factor1** | **Factor2** | **Uniqueness** | **Factor1** | **Factor2** | **Uniqueness** |
| RES | **0.7979** | −0.0205 | 0.3801 | 0.7937 | −0.0282 | 0.3917 |
| INVEST | **0.7138** | −0.0087 | 0.4970 | 0.7188 | 0.0236 | 0.4998 |
| ZONE | **0.5294** | −0.0746 | 0.7558 | 0.5479 | −0.0990 | 0.7442 |
| BRAND | **0.5378** | 0.1453 | 0.6073 | 0.4951 | 0.1779 | 0.6351 |
| STAFF | **0.7126** | 0.0742 | 0.4309 | 0.6419 | 0.1435 | 0.4753 |
| PRESS_TECH | 0.0352 | **0.2270** | *0.9388* | | | |
| STAB_TECH | 0.0170 | **0.5569** | 0.6796 | 0.0394 | 0.5358 | 0.6903 |
| GOAL | 0.0120 | **0.6034** | 0.6282 | 0.0083 | 0.6090 | 0.6240 |
| INDEX | 0.0389 | **0.6086** | 0.6032 | −0.0063 | 0.6499 | 0.5817 |
| EXP | 0.0182 | **0.6204** | 0.6029 | 0.0253 | **0.5987** | 0.6257 |

(*) In bold the highest loading for each indicator is highlighted. (**) In bold and italic loadings with uniqueness value larger than 0.8.

Results of the second FA, after dropping PRESS_TECH, show that three variables maintain communalities values larger than 0.5 (RES, INVEST, STAFF), one variable shows a value larger than 0.4 (INDEX) and the remaining variables have value of variance shared with other variables largely higher than 0.2 (GOAL, EXP, BRAND, STAB_TECH and ZONE). Moreover, the Kaiser–Meyer–Olkin (KMO) measure of sampling adequacy shows a value of 0.7454, meaning that FA is adequate.

Our estimates reveal that loadings are all highly acceptable, since there is no variable with values lower than 0.5. It follows that all the indicators are relevant in defining the factor's dimensionality. Moreover, at least four indicators are included for each factor, according to the general rule proposed by Fabrigar et al. [42]. Factor 1 is composed by indicators that exclusively refer to sources of asset specificities such as the presence of specific physical resources (RES), dedicated investments (INVEST), geographical specificity (ZONE), commercial brands (BRAND) and high-skilled workers (STAFF): as a result, it can be interpreted as a construct related to the (perceived) relevance of asset specificity. Factor 2 highlights that indicators related to the importance of both technological and behavioral aspects affect the level of uncertainty at stake. In more detail, indicators encompass technological stability (STAB_TECH), contractual transparency (GOAL and INDEX) and the importance of prior experiences (EXP). It follows that the second construct refers to the (perceived) relevance of the absence of uncertainty.

Based on the outcome of the first stage of the empirical analysis, factors were used to perform the econometric estimation. Henceforth, the results of the second stage of analysis are reported.

First, postestimation statistics show an overall rate of correct classification (cutoff value of 0.5) of 68.63%, with 87.14% of the normal weight group correctly classified (specificity) and only 28.13% of the low weight group correctly classified (sensitivity). This result implies that the classification is sensitive to the relative sizes of each component group. Regrouping data in 10 nearly equal-sized groups the Hosmer–Lemeshow goodness-of-fit highlights that the model fits reasonably well (*p*-value 0.725). The area under the ROC curve of 0.739 indicates acceptable discrimination for the model.

Table 3 shows both the coefficients of the probit models and marginal effects. Model A presents the estimates only on control variables. Model B includes the latent variables regarding our conceptual framework.

**Table 3.** Probit regression with robust errors: coefficients and marginal effects (m.e.).

| Variables | A | | | | | | B | | | | | |
|---|---|---|---|---|---|---|---|---|---|---|---|---|
| | Probit Model | | | m.e. | | | Probit Model | | | m.e | | |
| | Coeff | Z | $p > |z|$ | dy/dx | Z | $p > |z|$ | Coeff | Z | $p > |z|$ | dy/dx | Z | $p > |z|$ |
| Factor 1 | | | | | | | −0.201 | −1.27 | | −0.060 | −1.27 | |
| Factor 2 | | | | | | | 0.562 | 3.29 | ** | 0.168 | 3.57 | *** |
| AGE_2 | −0.401 | 0.396 | | −0.117 | 0.141 | | −0.659 | −1.29 | | −0.217 | −1.24 | |
| AGE_3 | −0.125 | 0.405 | | −0.033 | 0.146 | | −0.654 | −1.22 | | −0.215 | −1.19 | |
| EDU_LEV_2 | 5.196 | 0.298 | *** | 1.962 | 0.110 | *** | 4.324 | 7.92 | *** | 1.290 | 6.36 | *** |
| EDU_LEV_3 | 4.970 | 0.220 | *** | 1.884 | 0.095 | *** | 4.584 | 12.30 | *** | 1.375 | 8.33 | *** |
| EDU_LEV_4 | 5.057 | 0.299 | *** | 1.891 | 0.120 | *** | 4.846 | 12.21 | *** | 1.453 | 9.06 | *** |
| _constant | −4.977 | 0.421 | *** | | | | −6.134 | −5.47 | | | | |
| N. obs. | 150 | | | | | | 102 | | | | | |
| Log pseudolikelihood | −98.431 | | | | | | −54.402 | | | | | |
| Wald chi2 | 1007.18 *** | | | | | | 404.48 *** | | | | | |
| Pseudo R2 | 0.028 | | | | | | 0.142 | | | | | |

*** $p < 0.001$; ** $p < 0.05$; * $p < 0.10$.

Model B reveals that asset specificity (factor 1) does not play any significant role in affecting the use of production farming agreements in our case. However, our results show the relevant role of the absence of uncertainty (embedded in factor 2) in fostering the adoption of contracts in the durum wheat sector (m.e. 168). As for the control variables, whereas farmers' age does not play any role in the decision to adopt contracts, results

highlight that the higher is the education level of the farmers, the more they are likely to use contracts, compared to the less educated ones (m.e. vary from 1.290 to 1.453).

## 6. Discussion

Results reveal that sources of asset specificity related to food quality strategies in the durum wheat sector do not play a significant role in increasing the likelihood of adopting contracts. Therefore, since the impact of asset specificity on using contracts cannot be verified, the first hypothesis must be rejected. However, Allen and Lueck [43] noticed that in presence of quality-related specialized assets, usually idiosyncratic, contracts tend to be adopted. In the case under analysis, a possible explanation of the results lies in the nature of the durum wheat, to be considered as a staple crop. According to Hellin et al. [44], Bidzakin et al. [45] and Maertens and Valde Velde [46], since these types of products do not require a large amount of specific investments, then spot market is likely to represent a more attractive organizational solution to govern transactions. These findings, therefore, confirm that contractual relationships do not develop in a staple chain, such as the durum wheat, also in presence of food quality goals, unless (national or supranational) public authorities intervene, like in the case of the "Fondo Grano Duro" in Italy and the regulation on the Common Market Organization in EU, both paving the road for coordination solutions such as formalized written contracts [1,26].

As for the second hypothesis, our results highlight that the (absence of) uncertainty related to food quality strategies is able to increase the likelihood to sign a production contract in the durum wheat sector. Accordingly, the second hypothesis is supported. This finding reveals this type of contract is growing as a solution to mitigate uncertainty related to the relationships between farmers and buyers in the agri-food industry [10]. What emerges from our empirical analysis, therefore, confirms that production contracts can be seen as organizational responses stimulated by the uncertainty related to food quality strategies [22,47]. As highlighted by Fischer and Wollni [48], modern chains are characterized by a disaggregation of quality controls that implies high levels of uncertainty for farmers and buyers, leaving room for opportunistic behavior [5]. Production contracts, therefore, represent a solution for addressing uncertainty when they are properly designed and negotiated so as to foster clarity referred to quality standards, reciprocal experiences, and goals transparency, as well as to stabilize technological uptake [49]. In this regard, it is interesting to note that a large share of the sampled farmers using production contracts joined a cooperative. Such an interesting trend also emerged in the durum wheat sector when the likelihood of not conforming to requirements (and of incurring post-harvest-price discount if quality standards is not met) leads to contractual solutions aimed to reduce and stabilize uncertainty [14]. These, in turn, help to create a good climate of mutual trust that reduces maladaptation costs and allows agents to make contracts more stable and less expensive to enforce [50].

Moreover, the importance of transparency and trust is further supported by studies that investigated farmers' preferences for contract farming with choice experimental approaches [22,51]. Examples of farmers found to lack information about contract details or unaware of input prices, contract conditions and even of the exact company they signed the contract are manifold in the literature [35,52]. Written contracts can be problematic in particular, as they may lack transparency when using legal terms or language that is inaccessible to farmers with relatively low education levels, so that farmers are neither able to read nor to fully understand the contract prior to signing it [53,54]. Since the pasta supply chain in Italy has been also characterized by anticompetitive and unfair trading practices [55], the results even more underline the importance of transparent quality grading and trust for contract participation [48,56]. As can be seen from the results, it follows that the trustworthiness and reliability are stronger determinants for farmers' participation in contract schemes than the only price offered [57].

## 7. Conclusions

The present paper provides empirical evidences of the drivers of the adoption of production contracts in the durum wheat supply chain in Italy. We investigated the effects of asset specificity and uncertainty on the decision to adopt production contracts within a framework characterized by an increasing role of food quality. Using the reduced-form test of transaction costs, quantitative evidence contributes to confirming the negligible role of asset specificities in presence of a staple commodity characterized by low perishability and low investments. On the other hand, our study contributes in scrutinizing a traditional TCE hypothesis, highlighting that uncertainty also deserves attention in staples supply chains relationships, due to the growing sensibility of consumers for credence attributes of food related to its origin and quality. Long-term experience and sharing of clear and transparent objectives and strategies on quality standards allow to handle uncertainty, mainly reducing behavioral and technological risks and triggering the diffusion of production contracts. Our findings also bring some relevant implications, both for managers and policymakers. The former should consider farmers and buyers' preferences when designing contracts for staple crops, aiming to share clear and transparent strategies around quality goals. Better tailoring contracts can help, on the one hand, foster recursive interactions and strengthen reciprocal experiences and, on the other hand, improve participation rates to production contracts, paving the road for mutual gains. Policymakers should incentivize the spread of contractual arrangements aimed to ensure the provision of staple crops with enhanced qualitative characteristics, provided that both farmers and buyers' preferences are considered in contract design. This latter finding could be an important insight for the development of policies aiming to create more stable and long lasting relationships along the agri-food supply chain, and also in presence of low and decreasing level of uncertainty.

This study presents some limitations: First, the analysis offers scarce evidence on the comparison of the use of contracts in different organizational arrangements, such as spot market or cooperatives. Second, the reduced-form estimates adopted in the experimental part do not permit the identification of the structural relations that underlie the effects of the attributes. Furthermore, our results leave open the question related to elucidating the costs of mistaken coordination.

**Supplementary Materials:** The following are available online at https://www.mdpi.com/2071-1050/13/5/2921/s1, Annex S1: questionnaire, Annex S2: dataset.

**Author Contributions:** Conceptualization, S.C., G.M. and A.F.; methodology, S.C.; validation, A.F., G.M. and G.M.d.O.; investigation, A.F. and G.C.; resources, A.F.; data curation, S.C. and G.C.; writing—original draft preparation, S.C.; writing—review and editing, A.F., G.M., G.M.d.O. and G.C.; visualization, S.C. and A.F.; supervision, G.M. and A.F. All authors have read and agreed to the published version of the manuscript.

**Funding:** This research received no external funding.

**Institutional Review Board Statement:** Not applicable.

**Informed Consent Statement:** Not applicable.

**Data Availability Statement:** The data presented in this study are available in Supplementary material (Annex S2).

**Conflicts of Interest:** The authors declare no conflict of interest.

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
