# Peer review of "Production Contracts and Food Quality: A Transaction Cost Analysis for the Italian Durum Wheat Sector"

_sustainability, doi:10.3390/su13052921_

Round 1

Reviewer 1 Report

Research design, methodologies, and conclusion are clear.

However, there remain some minor revisions as follows.

-It is uneasy to distinguish Figure 1 since the color is similar.  

- Data description table can be added on.

- In page four, no description about "second section".

- Methodology of the probit model is obscure.  Explanation more in detail will be helpful for understanding. 

Author Response

COMMENTS

REPLY

It is uneasy to distinguish Figure 1 since the color is similar.  

We fully agreed. We edited the figure accordingly, so that colours can be easily identified.

Data description table can be added on.

According to your suggestion, we inserted the table from the Appendix in the main text (see Table 1).

In page four, no description about "second section".

According to your suggestion, we inserted a short description of the second section of the questionnaire.

Methodology of the probit model is obscure.  Explanation more in detail will be helpful for understanding. 

We added some more information to clarify  our probit  estimates.

Reviewer 2 Report

The scientific quality of the paper is good. However, I have a few minor comments. The abstract should be improved to better inform about the content of the paper: the research purpose, method, materials and results. In the "Discussion" section, based on the literature it would be valuable to consider the external conditions more broadly. For example, it is worth considering (if possible): How does the asymmetry of the position of producers and customers affect signing the contracts? Is the impact of imports also important? Is there a cyclicality here? Do CAP regulations matter?
The comments of the reviewer do not diminish the cognitive value of the paper, they are only suggestions for better perception by the readers.

Author Response

COMMENTS

REPLY

English language and style are fine/minor spell check required

Based on your feedback, we checked for language and style in order improve readability of the paper.

The abstract should be improved to better inform about the content of the paper: the research purpose, method, materials and results.

Thanks for your feedback. We changed the abstract according to your suggestions aiming to make it more informative for readers.

In the "Discussion" section, based on the literature it would be valuable to consider the external conditions more broadly. For example, it is worth considering (if possible):

How does the asymmetry of the position of producers and customers affect signing the contracts?

Is the impact of imports also important?

Is there a cyclicality here?

Do CAP regulations matter?

Thanks for this valuable suggestion. We tried to consider some of the suggested external conditions in the Discussion section. In more detail, we mentioned the role of the Common Market Organization within the CAP and we signalled the presence of unfair trading practices in the pasta sector.